# Popcorn Appearance of Severely Calcified Uterine Leiomyoma: Image-Pathological Correlation

**DOI:** 10.3390/diagnostics13010154

**Published:** 2023-01-02

**Authors:** Charuwan Tantipalakorn, Surapan Khunamornpong, Sirinart Sirilert, Theera Tongsong

**Affiliations:** 1Department of Obstetrics and Gynecology, Faculty of Medicine, Chiang Mai University, Chiang Mai 50200, Thailand; 2Department of Pathology, Faculty of Medicine, Chiang Mai University, Chiang Mai 50200, Thailand

**Keywords:** calcified leiomyoma, CT scan, postmenopausal woman, subserous leiomyoma, ultrasound

## Abstract

Calcified subserous leiomyoma is a rare benign tumor commonly seen in the postmenopausal age group. Cases with severely calcified degeneration all over the mass are extremely rare. It causes diagnostic confusion with the solid calcified adnexal mass and the large bladder calculi in the pelvis. We hereby present a case of heavily calcified subserous uterine leiomyoma in a 66-year-old postmenopausal woman. An X-ray of the abdomen and pelvis and CT scan showed a pelvic mass with scattered popcorn appearance in the pelvis, representing severely calcified discrete spots all over the mass. Sonographically, different from typical uterine leiomyomas which exhibit recurrent refractory shadowing patterns, our case showed heavy homogeneous acoustic shadow obscuring all structures beneath the mass surface, resulting in a suboptimal ultrasound examination. Accordingly, CT scans, which are usually not a primary tool for the diagnosis of uterine leiomyomas, are helpful to characterize the mass and identify their organ of origin. The case presented here was treated with a hysterectomy with bilateral oophorectomy and was post-operatively confirmed for severely calcified subserous leiomyomas.

Uterine leiomyoma is the most common tumor of the female pelvic organs, mostly occurring during reproductive age. The prevalence ranges from 4% in women 20–30 years of age to 11–18% in women 30–40 years of age and 33% in women 40–60 years of age [1]. In general, the diagnosis is based on ultrasound examination, which exhibits recurrent refractory shadowing pattern or multiple discrete shadows originating from within the mass [2]. However, it may be difficult to make the diagnosis in unusual cases of atypical findings. Subserous leiomyoma can simulate the adnexal mass, possibly mistaken for the ovarian tumor/malignancy, and exhibit different degenerative changes such as hyaline degeneration, cystic degeneration, fatty degeneration, red degeneration, and calcified degeneration. Uterine leiomyoma with calcified degeneration is rather uncommon, but severely calcified degeneration all over the mass is extremely rare and has been reported in a very limited number of cases [3,4,5,6]. Here, we describe a severely calcified leiomyoma forming a scattered popcorn pattern in a postmenopausal woman. It can lead to some diagnostic confusion with ovarian teratoma/malignancy or large bladder calculus in the imaging investigation. Sonographically, different from typical uterine leiomyomas, our case showed heavy homogeneous acoustic shadow obscuring all structures beneath the mass surface, resulting in a suboptimal ultrasound examination. Accordingly, CT scans, which are usually not a primary tool for the diagnosis of uterine leiomyomas, are helpful to characterize the mass and identify their organ of origin. The case presented here was treated with a hysterectomy with bilateral oophorectomy and was post-operatively confirmed for severely calcified leiomyomas. The objective of this report is to describe the unusual appearance of leiomyoma which might be mistaken for adnexal mass and focus on the limitation of ultrasound examination as well as the usefulness of CT scans.

Calcification tends to develop in uterine leiomyomas in the absence of vascular supply, especially in postmenopausal women. Calcification of leiomyomas may present in various patterns such as a mottled or popcorn appearance with no well-defined curvilinear outline, totally calcified as a solid stony mass or a well-defined outer part, or the high-attenuation border with less calcification of the inner part. The presence of calcification in the uterus is a reliable sign of uterine leiomyoma [7]. Nevertheless, calcified leiomyoma is rather uncommon. The prevalence of calcified uterine leiomyomas is reported to be 3–10% of cases [3]. However, in a majority of cases, calcifications are confined to only a small part of leiomyomas. Extensive calcification involving all of the mass as seen in our case is extremely rare.

Though uterine leiomyomas can be recognized with confidence by ultrasound in most cases, large leiomyoma with various types of degeneration can be a diagnostic challenge. Moreover, they can be mistaken for ovarian tumors or malignancy, especially in postmenopausal women [8]. An ultrasound examination of severely calcified mass is less informative. Generally, a diagnosis of a pedunculated subserous leiomyoma can be made by sonographic visualization of a vascular pedicle connecting to the uterus. Nevertheless, such a finding may not always be demonstrated by ultrasound [2], and it is very difficult or impossible in cases with severe calcifications as seen in our cases because of heavy acoustic shadow. Accordingly, in these cases, a CT scan is helpful to characterize pelvic calcified masses and identify their organ of origin [9], though CT scans are usually not the primary tool for the diagnosis of leiomyomas.

Typical leiomyomas are associated with the presence of three or more well-defined shadows originating from within a mass, defined as the recurrent refractory shadowing pattern [2,10], not originating from echogenic lead points which are presumed to arise from calcifications. Pathologically, these discrete recurrent shadows originate from transitional zones between the margins of smooth muscle whorls and the margins of fibrous connective tissue within the leiomyoma [10]. The ultrasound beam is refracted and distorted when it passes through the different tissues or curved surfaces. This pattern is typically expressed by uterine leiomyomas and is used as a diagnostic sonographic sign. The heavy shadowing in our case was not a recurrent refractory shadow pattern of leiomyoma but an acoustic shadow, which occurred because no ultrasound beams passed through the multiple layers of calcifications.

Figure 1 and Figure 2 illustrate the discrepancy in image quality between the two modalities. The stacked calcific pellets in numerous layers produced stacked shadows on conventional ultrasound to completely obscure the underlying structures. The numerous calcific pellets, proven by pathological examination in Figure 3 and Figure 4, corresponded to the numerous popcorn spots on the pre-operative CT scan image (Figure 1), whereas they could not be correlated with the ultrasound findings (Figure 2). In conclusion, our findings emphasize that, though leiomyoma can usually be diagnosed with conventional ultrasound without difficulty, in cases of severe calcifications, it is difficult to make a diagnosis by ultrasound because of severe acoustic shadows leading to a non-informative examination. In this circumstance, a CT scan is superior, as it illustrates a clear and typical popcorn appearance of heavy calcifications. Accordingly, in cases of non-informative ultrasound, which is used as a primary tool of pelvic imaging, a CT scan should be performed to make the diagnosis.

## Figures and Tables

**Figure 1 diagnostics-13-00154-f001:**
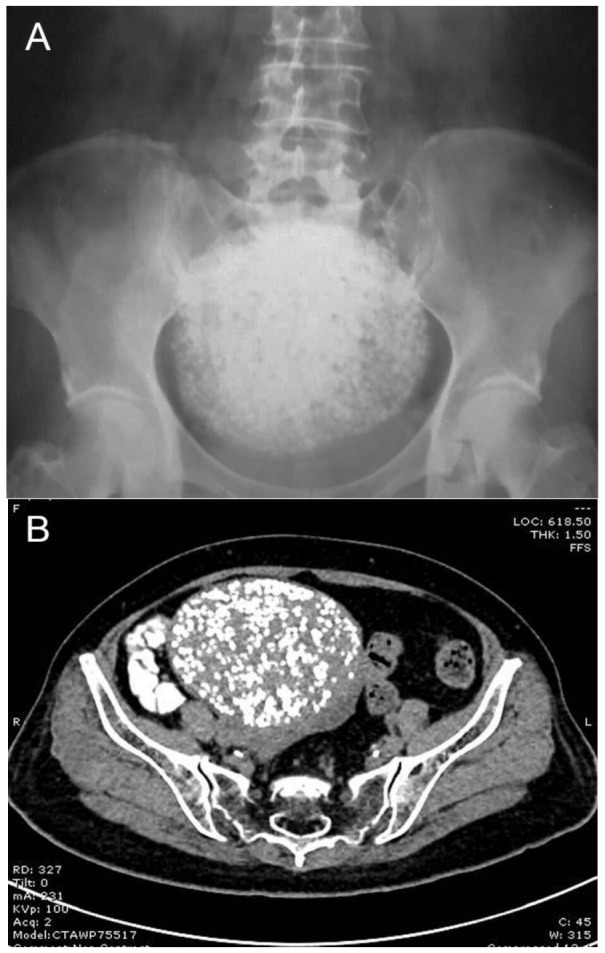
A 66-year-old nulliparous, postmenopausal Thai woman presented with lower back pain as well as pelvic discomfort for two weeks. Initial physical examination revealed a pelvic mass, no tenderness, a relatively smooth surface, firm consistency, and an estimated size of approximately 16–18 weeks of pregnancy. A plain film X-ray of the abdomen was performed to rule out orthopedic disorders because of low back pain symptoms. The X-ray findings showed a pelvic mass with severe calcification giving a scattered popcorn appearance, as presented in (**A**), whereas other bony structures were within normal limits. She was referred to our hospital for further management. Transabdominal ultrasound of the pelvis showed a mass with heavy acoustic shadow, resulting in non-informative details of the mass, and no vascularization. Because of the suboptimal ultrasound examination, a CT scan was performed and revealed an unusual pattern of popcorn appearance in the pelvis, as presented in (**B**). Severely calcified subserous leiomyoma uterus was a provisional diagnosis. The popcorn appearance on the cross-sectional plane of the CT scan of the pelvis (**B**) is much more clearly demonstrated than the AP plane of the film X-ray of the abdomen (**A**). Note that the mass was connected to the uterus. The mass measured 10.8 × 8.7 × 9.8 cm in diameter. The numerous small discrete calcified pellets were densely scattered all over the mass, giving the popcorn appearance, different from a homogeneous stony mass. The differential diagnoses of severely calcified mass may include ovarian teratoma, bladder calculi, calcified uterine leiomyoma, and bone tumors. A 3.7 × 2.9 cm right ovarian cyst was also noted. The left ovary appeared normal. The urinary bladder was compressed by an enlarged mass. No ascites and lymph node enlargement were seen.

**Figure 2 diagnostics-13-00154-f002:**
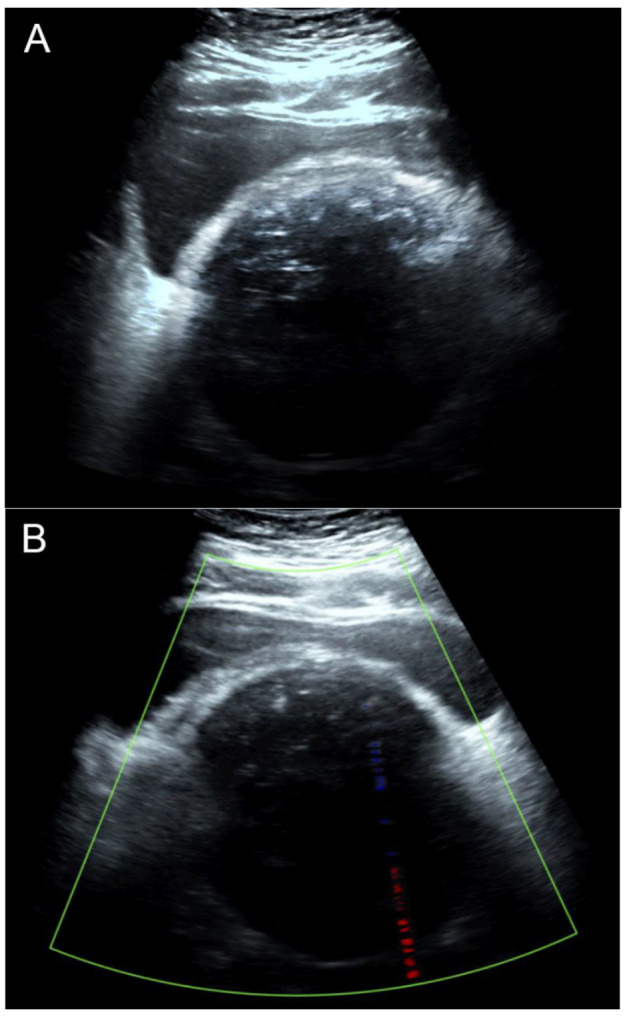
Transabdominal ultrasound images: (**A**) The 2D ultrasound showed a pelvic mass of uncertain anatomical origin, with severe acoustic shadow, obscuring all details behind the calcifications on the surface of the mass. (**B**) Color flow mapping showed no vascularization. Note that there were no multiple-edge shadows typical for uterine leiomyomas because multiple calcifications completely obscured all of the structures underneath the surface facing the ultrasound beam. The ultrasound beam cannot pass through several layers of calcific discrete pellets. Accordingly, the popcorn appearance cannot be produced by ultrasound, and the examination should be considered suboptimal and non-informative. Though uterine leiomyomas can sonographically be diagnosed with confidence in most cases, severely calcified leiomyomas can be a diagnostic challenge because of suboptimal and non-informative examinations.

**Figure 3 diagnostics-13-00154-f003:**
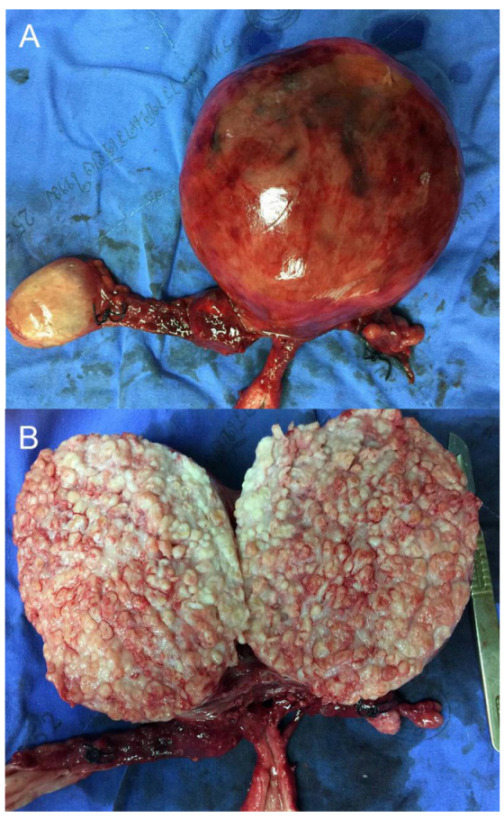
Total abdominal hysterectomy and bilateral salpingo-oophorectomy were performed. Post-operative findings: (**A**) The gross appearance of the sub-serous pedunculated leiomyoma; (**B**) The cut-surface of the mass showed numerous small discrete calcified pellets, which were densely packed all over the mass. The numerous calcified pellets are consistent with a scattered popcorn appearance on the imaging presented in Figure 1.

**Figure 4 diagnostics-13-00154-f004:**
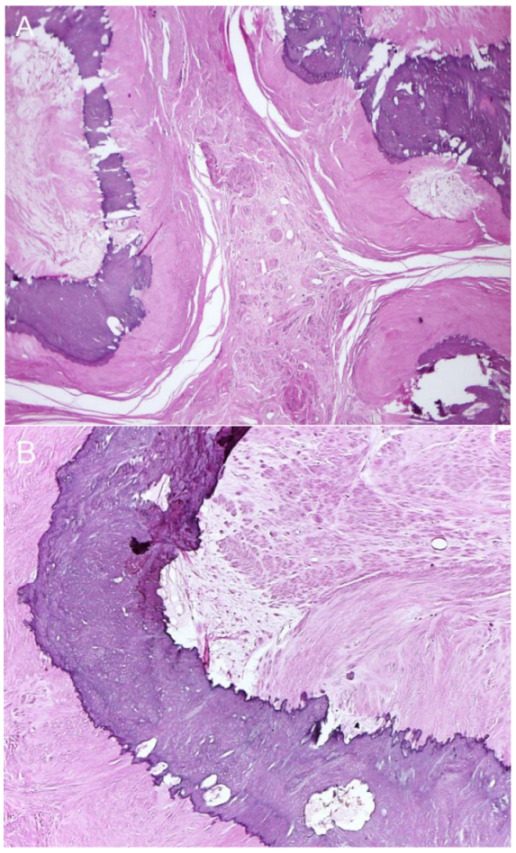
Operative findings and pathological examination confirmed a severely calcified uterine leiomyoma. The mass was well demarcated and composed of multiple nodules of collagen-rich smooth muscle tissue, which contained frequent central islands or irregular ring-like bands of calcifications. The smooth muscle cells were arranged in interlacing fascicles and showed bland and uniform nuclei, without atypia or increased mitotic activity. Microscopic pathology: (**A**) Collagen-rich nodules of smooth muscle tissue containing calcifications in the central zone. (**B**) Interlacing fascicles of smooth muscle cells without nuclear atypia adjacent to the calcification band.

## Data Availability

The data of this report are available from the corresponding authors upon request.

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
