# Peer review of "Popcorn Appearance of Severely Calcified Uterine Leiomyoma: Image-Pathological Correlation"

_diagnostics, 2023, doi:10.3390/diagnostics13010154_

Round 1
Reviewer 1 Report
I enjoyed reading the report very much. I have several remarks that require your attention:
1. The level of english is average and requires extensive proofing before the manuscript may be accepted.
2. The legends for figures 1-4 are too long. They should be short and concise and describe the specific findings and not tell a story.
3. The US images are of extremely low quality and do not reflect the level of expertise I would expect from a proffesional US. Either replace with higher quality images or consider removing the image. The rest of the images in the manuscript are excellent.
4. The structure of the whole manuscript is lacking. There is a lot of repetition that is unnecasary. I recommend not repeating the information in the body of the text in the figure legends.
5. Furthermore, I recommend a concise introduction regarding the incidence of fibroid formation, diagnosis of fibroids, complications assoicated with fibroids and treatment.
6. The case description is lacking. You need to describe the patient`s basic information, clinical presentation, circumstances of diagnosis of the fibroid, clinical considerations and differential diagnosis, imaging, blood work if done (CA-125 level possibley) and the treatment.
7. The description and discussion should discuss the difficulties in diagnosis, possible superior imaging modalities in these circumstances and options for reaching an accurate diagnosis.
Author Response
Reviewer 1
Comments and Suggestions for Authors
I enjoyed reading the report very much. I have several remarks that require your attention:
- The level of English is average and requires extensive proofing before the manuscript may be accepted.
Response: English is now checked and edited by the professional English editing service, as indicated by the attached certificate.
- The legends for figures 1-4 are too long. They should be short and concise and describe the specific findings and not tell a story.
Response: Because this report is an “Interesting Images” type, which we can add any helpful information to legend which can be long and added citations. The article type does not allow us to have “Heading” like “Discussion” or “Case presentation” etc. while case description and associated information / knowledge can be added to the figure legends as much as needed. Therefore, we make a request to keep the format as the way it is. However, if the Editor or the Reviewer strongly suggests to change the article type to be “Case Report”, we are willing to comply.
- The US images are of extremely low quality and do not reflect the level of expertise I would expect from a proffessional US. Either replace with higher quality images or consider removing the image. The rest of the images in the manuscript are excellent.
Response: That is the main purpose of the article to show that even with high resolution US, the image quality is still very poor because of the calcified tumor. We want to point out that US is less useful in the case like this and need CT or MRI. We want to illustrate the limitation of US in this case. The image is poor but it is the main point of this case to show the much different from excellent illustration on CT scans. Accordingly, we make a small request to keep the US images and no other images are better than this. All US images in this case are poor because of tumor nature, not because of author’s technique. Poor US image (non-informative US) and excellent CT images are the main insight we want to present.
- The structure of the whole manuscript is lacking. There is a lot of repetition that is unnecasary. I recommend not repeating the information in the body of the text in the figure legends.
Response: Because this report is an “Interesting Images” type, which need no structure. The repeating information is deleted. Originally, we submit as case report with structured style. However, on first submission, the editor suggested us to change to be “Interesting Image” type.
- Furthermore, I recommend a concise introduction regarding the incidence of fibroid formation, diagnosis of fibroids, complications assoicated with fibroids and treatment.
Response: Again, “Interesting Images” type, needs no “Introduction”. However, in the revised MS, we add some concise introduction in the first paragraph. However, fibroid treatment is out of scope of this presentation. We kindly request not to add the treatment.
- The case description is lacking. You need to describe the patient`s basic information, clinical presentation, circumstances of diagnosis of the fibroid, clinical considerations and differential diagnosis, imaging, blood work if done (CA-125 level possibley) and the treatment.
Response: Again, “Interesting Images” type, needs no part of case description but focusing on the images. However, we add some case description in the legend.
- The description and discussion should discuss the difficulties in diagnosis, possible superior imaging modalities in these circumstances and options for reaching an accurate diagnosis.
Response: In the revised MS, the difficulties in diagnosis, possible superior imaging modalities in these circumstances are discussed as suggested, as highlighted.

Reviewer 2 Report
In this manuscript, Tantipalakorn et al. reported a case of leiomyoma with severe calcified degeneration. The figure clearly supports author's conclusion, which helps with the diagnosis in clinic. CT scan would be a considerable tool to identify the origin of the tumor and avoid misdiagnosis. The work is of guiding significance in clinic.
Author Response
Thank you very much for the encouraging comment.
English proof-editing certificate is attached.

Reviewer 3 Report
no comments
Author Response
Response: no comment
English proof-editing certificate is attached.

Reviewer 4 Report
discussion need to be more accurate
Author Response
Comments and Suggestions for Authors
discussion need to be more accurate
Response: We add some discussion (at the end of article) to be more accurate as suggested, as highlighted.
Round 2
Reviewer 1 Report
Your reply is satisfactory. thank you for the clarifications.
Good luck.